# *Salmonella* spp. in Aquaculture: An Exploratory Analysis (Integrative Review) of Microbiological Diagnoses between 2000 and 2020

**DOI:** 10.3390/ani13010027

**Published:** 2022-12-21

**Authors:** Yuri Duarte Porto, Fabiola Helena dos Santos Fogaça, Adriana Oliveira Andrade, Luciana Kimie Savay da Silva, Janine Passos Lima, Jorge Luiz da Silva, Bruno Serpa Vieira, Adelino Cunha Neto, Eduardo Eustáquio de Souza Figueiredo, Wagner de Souza Tassinari

**Affiliations:** 1Department of Animal Parasitology, Institute of Veterinary, Federal Rural University of Rio de Janeiro (UFRRJ), Seropédica 23897-000, Brazil; 2Brazilian Agricultural Research Corporation, Embrapa Agroindústria de Alimentos, Rio de Janeiro 23020-470, Brazil; 3Department of Mathematics, Institute of Exact Sciences, Federal Rural University of Rio de Janeiro (UFRRJ), Seropédica 23897-000, Brazil; 4Department of Food and Nutrition, Federal University of Mato Grosso (UFMT), Cuiabá 78060-900, Brazil; 5Federal Institute of Education, Science and Technology of Mato Grosso (IFMT), São Vicente da Serra 78106-000, Brazil; 6Department of Veterinary Medicine, Federal University of Uberlândia (UFU), Uberlândia 38410-337, Brazil

**Keywords:** *Salmonella*, fish farming, food safety, public health, word cloud, similarity, Reinert’s algorithm

## Abstract

**Simple Summary:**

Salmonellosis is characterized by a gastrointestinal infection resulting from the ingestion of water and food contaminated by bacteria of the genus *Salmonella*, of which causes enterocolitis. Although infections are acute and self-limiting, efforts to prevent this problem of interest to global public health are important. *Salmonella* spp. is distributed in different environments and animal species; therefore, meat derived from animal production is one of the main routes of human infection, increasing the importance of research focusing on microbiological quality and food safety. Aquaculture is a constantly growing sector in the world, and the monitoring of *Salmonella* spp. in fish products is important for public health due to the risks of contamination during all stages of production. In this context, the present study carried out a systematic integrative review of the microbiological diagnoses of *Salmonella* spp. in aquaculture between 2000 and 2020 with the objective of characterizing and contributing to the promotion of measures to control and prevent this pathogen in aquaculture production. A database generated was composed of information that was mined from articles such as the most sampled aquaculture species, the microbiological diagnostic method(s) conducted in the investigation of *Salmonella* spp., and the main reported serotypes.

**Abstract:**

The present study aimed to characterize, through descriptive statistics, data from scientific articles selected in a systematic integrative review that performed a microbiological diagnosis of *Salmonella* spp. in aquaculture. Data were obtained from research articles published in the BVS, Scielo, Science Direct, Scopus and Web of Science databases. The selected studies were published between 2000 and 2020 on samples of aquaculture animal production (fish, shrimp, bivalve mollusks, and other crustaceans) and environmental samples of aquaculture activity (farming water, soil, and sediments). After applying the exclusion criteria, 80 articles were selected. Data such as country of origin, categories of fish investigated, methods of microbiological diagnosis of *Salmonella* spp., sample units analyzed and most reported serovars were mined. A textual analysis of the word cloud and by similarity and descending hierarchical classification with the application of Reinert’s algorithm was performed using R^®^ and Iramuteq^®^ software. The results showed that a higher percentage of the selected articles came from Asian countries (38.75%). Fish was the most sampled category, and the units of analysis of the culture water, muscle and intestine were more positive. The culture isolation method is the most widespread, supported by more accurate techniques such as PCR. The most prevalent *Salmonella* serovars reported were *S*. Typhimurium, *S*. Weltevreden and *S*. Newport. The textual analysis showed a strong association of the terms “*Salmonella*”, “fish” and “water”, and the highest hierarchical class grouped 25.4% of the associated text segments, such as “aquaculture”, “food” and “public health”. The information produced characterizes the occurrence of *Salmonella* spp. in the aquaculture sector, providing an overview of recent years. Future research focusing on strategies for the control and prevention of *Salmonella* spp. in fish production are necessary and should be encouraged.

## 1. Introduction

The genus *Salmonella* spp. belongs to the Enterobacteriaceae family and has important metabolic characteristics that not only promote its survival in the gastrointestinal tract environment but also have virulence mechanisms to escape defense cells, reproduce and cause homeostatic disturbance in the host [1,2]. *Salmonella* spp. causes salmonellosis, a classic infection characterized by enterocolitis, which in most cases is acute and self-limiting, making diagnoses difficult and resulting in underreporting and compromising actions to control and prevent new outbreaks.

Salmonellosis is a foodborne disease (FDA) and represents an important public health problem in several countries around the world [3]. Nontyphoid serovars of the genus *Salmonella* spp. are responsible for much of the incidence of foodborne outbreaks, with a variety of foods serving as vehicles for the occurrence of salmonellosis in humans [3,4,5]. Such foods include meat (beef, chicken, pork and fish), eggs, milk, cheese, fresh fruits, fruit juices, vegetables and fish [5,6,7].

*Salmonella* spp. is widely distributed in the environment and in several species of wild animals, production animals and pets. Mammals, fish, birds, reptiles, amphibians and plants can act as reservoirs and disseminators of *Salmonella* spp. [8,9,10]. However, the adoption of correct and strategic measures in livestock management can help to prevent and reduce the risk of contamination by *Salmonella* spp., since its spread can occur at several stages along the production chain, including all stages of production, processing, distribution, marketing and handling/preparation [11,12,13,14].

*Salmonella* spp. has been isolated from points in the fish production chain [1,15]. There is a consensus among researchers that *Salmonella* spp. does not naturally belong to the aquatic environment, although it is isolated from water, fish and products derived from aquaculture [15,16,17,18]. The contamination of fish by *Salmonella* spp. is little known, since research has reported that fish act as a host of the bacteria for relatively short periods of time without any description of symptomatic manifestation of the disease [15,16,17,18,19]. Thus, *Salmonella* spp. has been isolated not only from the viscera but also from the gills and skin of the fish, contributing to an increased risk of cross-contamination during handling, processing, storage and commercialization due to failures in hygienic-sanitary care or using equipment, surfaces and utensils that are inadequately sanitized during the production chain [12,20,21,22].

It is notable that the health of fish depends on the quality of the water. In this way, the chemical, physical and microbiological factors of water are extremely important. Among the possible sources of contamination, water appears to be a probable vehicle, as it is widely exposed to human, agricultural and industrial pollution [23,24,25]. In addition, the water used for aquaculture itself can be a source of contamination. Research has been carried out to assess the presence of *Salmonella* spp. in surface water and sea water and the subsequent contamination of fish products and aquaculture [11,26]. Fresh water is often contaminated by *Salmonella* spp. through effluents, and consequently, coastal waters; shellfish and fish farming areas are especially subject to this pathogen [26,27,28]. Even if the presence of *Salmonella* spp. in aquaculture production species is reported with low prevalence, it is a pathogen of public health interest, and the detection techniques in the prevalence must be observed, tested and improved.

Methods for the rapid and accurate identification of pathogens in the production chain are important for ensuring food quality and for taking control measures through traceability to minimize economic losses and damage to public health. The health risks associated with the consumption of low-quality aquaculture foods make the evaluation and control of food safety topics a global concern [1,7,29]. Microbiological controls must be addressed in designated aquaculture environments for proper production management practices and for consumer education programs.

In this context, an integrative review of investigations of *Salmonella* spp. in aquaculture between 2000 and 2020 was decided to be performed, with the influence of the systematic review model already observed in other reviews, as it systematizes the research procedures, making the method explicit and a more auditable and reproducible model to facilitate the consolidation of knowledge [30,31]. The selected articles aimed to characterize and produce information through descriptive statistics techniques and text mining [32,33]—this refers to mined data regarding the years of publication, country of origin, area of study, categories of most sampled aquaculture species, sampling origin, methods of microbiological diagnosis of *Salmonella* spp., most evaluated units of analysis and most identified *Salmonella* serovars to contribute to the control and prevention of *Salmonella* spp. in fish production. The textual analysis of the abstracts was performed with word cloud grouping of similarity and descending hierarchical classification with the application of Reinert’s algorithm [34,35]; this was performed to make it possible to identify, organize and classify the most predominant different themes addressed in scientific articles for a better interpretation of the addressed themes in investigations into the microbiological diagnoses of *Salmonella* spp. in aquaculture.

## 2. Materials and Methods

### 2.1. Data Sources and Research Strategy

The searches performed in the electronic bibliographic databases BVS, Scielo, Science Direct, Scopus and Web of Science were restricted to the key terms: (fish farming) AND (*Salmonella*) OR (salmonellosis); the period between 2000 and 2020; English and Portuguese; original article formats. A total of 3373 articles were identified (Figure 1).

### 2.2. Research Design

The integrative review was based on the steps established by the protocol for Systematic Reviews and Meta-Analyses (PRISMA) [36]. All the steps and strategies for selecting articles are detailed in Figure 1.

### 2.3. Selection Process

Two reviewers (YDP and AOA) performed the selection of possibly relevant articles by critically reading the titles and abstracts. Thus, 3197 articles were excluded, as they were considered to have no direct relationship with the subject studied. Zotero^®^ Reference Manager version 5.0.96.2. identified 60 duplicate articles that were excluded. A total of 116 articles were retrieved and selected. All disagreements were resolved by consensus.

### 2.4. Eligibility Criteria

Primary studies tested for *Salmonella* spp. with reports of positive or negative contamination from samples of species produced in aquaculture, environmental samples (culture water, soil and sediments) from the breeding phase or as a product and derivatives for sale in markets. Therefore, in the screening stage, the studies focused only on the resistance or activity of *Salmonella* spp. against antimicrobials (12), informative reviews (8), diagnosis of *Salmonella* spp. in water and other environmental samples not related to aquaculture (12) and studies without access to the full document (4). Thus, 80 productions were chosen for this review.

### 2.5. Abstract and Data Analysis

Two reviewers (YDP and WST) performed a critical analysis of the contents of the articles that were chosen for data extraction. Two reviewers (WST and AOA) verified the data consistency.

The information extracted was the year of publication, authors, research title, name of the published journal and country. Information such as sample origin, species studied in the investigation, sample rate(s) tested, sample size tested, diagnostic assay(s) performed, number of positive samples and serotypes were also extracted. Some information has been categorized for better interpretation.

All data analysis was performed using the R^®^ statistical package version 4.0.2 (22 June 2020) (R Foundation for Statistical Computing, Vienna, Austria) [37]. The libraries used were “tm” and “SnowballC” for text mining and text stemming [38,39], the “word cloud” generator [40], and the world map generator to analyze the frequency of productions by country [41]. The Iramuteq^®^ program (Interface de R pour les Analyses Multidimensionneles de Textes et de Questionnaires) (LERASS—Laboratoire d’Études et de Recherches Appliquées en Sciences Sociales, Toulouse, France), developed in R^®^, was used for the analysis of similarity and descending hierarchical classification (DHC) with the application of Reinert’s algorithm of the content of abstracts [34,35].

### 2.6. Textual Analysis

The use of classical techniques of exploratory data analysis [32] and analytical techniques of text mining [33] for the analysis of unstructured data identified patterns that characterized the production from the perspective of the presence of *Salmonella* spp. in aquaculture.

A database was generated containing the abstracts of the articles selected in the integrative review, generating a corpus with 17,142 words, in which a word cloud composed of the most frequent words, a similarity graph and a descending hierarchical classification (DHC) graph were produced. From the similarity analysis, it was possible to identify the intensity of the occurrence of the words and the indications of the connectedness between them. The similarity graph presents a tree structure with ramifications; therefore, the content of the selected abstracts can be characterized by the identification of the most used words and the proximity between them. To generate the graphic image that was more visually readable, cleaning was performed by cutting out some words of less or no importance with the context and selecting the terms whose frequencies were equal to or greater than 20. Finally, the descending hierarchical classification technique was applied (DHC) to generate a graph containing the textual corpus of the abstracts divided by groupings of text segments according to the similarity of the terms used. The objective of the entire textual analysis of the abstracts was to enable the identification of the most predominant distinct themes addressed in the scientific articles, organize them, and thus classify them for a better interpretation of the themes addressed in investigations on the microbiological diagnoses of *Salmonella* spp. in aquaculture.

## 3. Results

### 3.1. Number of Articles, Study Themes, Textual Analysis and Countries

The present review identified 3373 scientific articles through the search strategy adopted, where 80 research articles were chosen for samples investigated for the presence or absence of *Salmonella* spp. in the aquaculture production chain (Figure 1).

The articles were published by 47 different journals to address and disseminate the results obtained with a strong multidisciplinary character on *Salmonella* spp. as a contaminating pathogen in aquaculture. A comprehensive table summarizing all the essential information on the origin of studies, samples and diagnostic techniques used in each of the 80 included studies is available as Appendix A.

It was possible to categorize seven main areas of study that the authors used to disseminate their research, emphasizing the importance of surveillance of *Salmonella* spp. in food produced by aquaculture, and the interest in a microbiological health problem that touches different fields of knowledge. That is, it is considered an essentially multidisciplinary topic that can cover areas related to food science (38 articles), animal production (12 articles), environment (11 articles), microbiology (6 articles), veterinary (5 articles), health (2 articles) and multiple areas (6 articles).

Following the idea of a multidisciplinary approach, the topics of articles involving *Salmonella* spp. presented proposals that are diverse in objectives and studies. From the abstracts of the articles selected by the integrative review, it was possible to generate the word cloud (Figure 2) and the similarity graph (Figure 3) for textual analysis.

The word cloud (Figure 2) was formed by the most frequent words contained in the abstracts. The higher the frequency of the word, the larger the font size of the word represented in the cloud. In this analysis, the five most prominent words in decreasing order of frequency (n_i_) were “*Salmonella*” (212), “fish” (207), “resistance” (85), “bacteria” (80) and “water” (79), while the other words contained in the cloud had a frequency (n_i_) lower than 64. The greater use of these most prominent words indicates that the selected articles emphasized *Salmonella* spp. (“*Salmonella*”) as an important pathogen kept under surveillance and microbiological investigation in the aquaculture production chain. In this sense, the word “bacteria” can generally support interest in microbiological safety and safe food for products from aquaculture. The “fish” represented the category of fish most affected or of greatest interest for microbiological surveillance against *Salmonella* spp. that would pose risks of causing a DTA. The words “water” and “resistance” can represent, respectively, the vehicle that most contributes to the spread and silent contamination of the pathogen in the breeding phase and the concern about the characterization of the isolates regarding the susceptibility or resistance profile to antimicrobials.

For similarity analysis, the composition of the abstracts of the articles generated a graph (Figure 3) with the most frequent central words (n_i_) “fish” (166), “sample” (151), “isolate” (137), “*Salmonella*” (126) and “study” (95). The branches make links with other words that were often mentioned simultaneously in the abstracts, and therefore, their relationship with the respective central words is observed. In this way, it was possible to contextualize and understand how the theme of *Salmonella* spp. in aquaculture was found to be structured in the articles selected by the integrative review.

The co-occurrence of the words “fish”, “water”, “pond” and “bacterium” point to freshwater fish and water, respectively, as the category of fish and the medium that most favors microbiological dispersion; these were most frequently used and analyzed words for the microbiological diagnoses of *Salmonella* spp. The use of the word “sample” was frequently followed by words that denote the authors’ reports on the investigation, detection and identification of strains (“strain”) or species of *Salmonella* (“*S. enterica*”) in several samples of aquaculture species, characterized by the terms “seafood”, “shrimp” and “mussel”.

Next to the word “*Salmonella*_spp.” are “*Escherichia coli*” and “*Enterobacter* spp.”, representing the authors’ interest and reports in the investigation and microbiological diagnosis of other important bacterial pathogens belonging to the Enterobacteriaceae family in aquaculture species. Associated in this same context is the word “*Listeria monocytogenes*”, as this is another foodborne bacterial pathogen that causes serious problems of public health importance.

The minor ramifications associated with the term “study” characterize in a more detailed way the directions taken by the authors in relation to the object of investigation (*Salmonella*), as “aquaculture”, “farm” and “food” are referred to as the primary source of the investigation from livestock and food production, which together with the terms “prevalence” and “presence” reveal worrying results regarding food microbiological safety. This statement can be more evident when associated with the term “antimicrobial_resistance”, as this indicates that research has been carried out regarding the control of a chronic problem of contamination. In this way, the term “isolate” is related to biomolecular techniques based on the polymerase chain reaction (“PCR”) used in diagnostics, as well as bacterial isolates with an antimicrobial resistance profile or those submitted from susceptibility or resistance analysis to groups of antibiotics. “Tetracycline” was the most reported by the authors.

To help identify the themes addressed in the selected articles, and thus verify the privileged lines of studies, a graph was generated using the descending hierarchical classification (DHC) technique. Thus, five different classes or lines of study were obtained by grouping text segments with the greatest similarity in terms of the words used. Figure 4 and Figure 5 present the DHC results of the abstracts included in this review. The most representative class contained 25.4% of the text segments. The least representative class was only 13.9%.

In perspective, Figure 4 and Figure 5 provide a visualization of the terms in each class performed in the two-dimensional space graph so that it is possible to evaluate the composition of each class based on the positioning and intensity and on the size of the words. Note that while classes 4 and 5 are farther apart, each in a different quadrant of the plane, the others show some level of overlap (see Figure 4), which indicates that they share common contents. Based on the frequency of occurrence of terms in each class, it was possible to assign a name that summarizes the general meaning of each grouping of texts (Figure 5).

It was possible to verify that class 5 was separated from the others, which implies a greater differentiation of its content in relation to the other classes (Figure 5). Another division occurred with the separation of class 4, then further division into the three remaining classes occurred based on great similarity with each other.

Class 5, words such as “storage”, “sensory”, “shelf” and “quality” (Figure 4), stood out. This class was named “food safety”, containing terms that mention the period that the food is guaranteed to be free from pathogens or the action of spoiling microorganisms that would modify the quality, making it unfit for consumption. Further and different from class 5 is class 4, which was named “antimicrobials”, where the most frequent terms were “resistance”, “resistant”, “ampicillin” along with several other terms that are names of other antimicrobials. This greater differentiation occurred due to the studies focusing on testing how the isolated pathogens interact with the action of different antimicrobial agents.

Class 1 (“microorganisms”), class 2 (“one health”) and class 3 (“profiles”) have marked similarity to each other. The term “*aquaculture*” (cluster 2) is found together with the terms “food” and “public health”, indicating the importance of this production activity for human food in a sustainable way through the practice of cultivation, together with collective health policies. Terms referring to other microbiological hazards to be prevented in foods from aquaculture are found in class 1, for example, “*Staphylococcus aureus*”, “*Aeromonas* spp.” and “*Salmonella* spp.”, indicating other pathogens that are potentially investigated and isolated in research in different types of samples. Otherwise, class 3 grouped terms such as “multiple antibiotic resistance”, “prevalence” and “susceptibility”, indicating that it is representative of terms that refer to the occurrence and microbiological profile of the investigated isolates.

Aquaculture producers from 37 countries located on all continents, except for Oceania, gave rise to the scientific productions chosen by the integrative review (Figure 6). Therefore, it was possible to observe continuous surveillance of *Salmonella* spp. in the aquaculture sector through the publications analyzed over the last few years.

Among all the nations, Brazil was the country with the highest number of articles selected by the integrative review, representing 10% of the selected articles (8 articles). The countries of the Asian continent accounted for 38.75% of the publications (31 articles), and China [17,25,42,43,44,45,46], India [47,48,49,50,51,52,53], Malaysia [54,55,56], Vietnam [57,58,59] and Thailand [60,61] had the highest number of elected publications. The countries of the European continent contributed 25% (20 articles), the second highest percentage of selected publications, with Spain [27,62,63,64,65,66], Italy [67,68,69] and Germany [70,71] having greater prominence. On the American continent, with a percentage of 21.25% (17 articles), the largest number of selected publications is in Brazil [9,18,23,72,73,74,75,76], followed by the USA [14,77,78,79,80] and Chile [81,82], while on the African continent with 15% (12 articles), Egypt [83,84,85], Nigeria [12,86], Tunisia [87,88], Kenya [89,90] and Turkey [91,92] had the highest number of selected works. The nations that had 1 publication elected were Algeria [21], Bangladesh [93], Belgium [94], Cameroon [95], Colombia [96], Denmark [97], England [98], Ghana [28], Greece [99], Hungary [100], Iran [13], Iraq [101], Japan [102], Latvia [103], Lebanon [104], Mexico [105], Norway [106], Poland [107], Portugal [22], Saudi Arabia [108], and Zambia [109].

The frequency of scientific publications between 2000 and 2020 was verified (Figure 7). An increase in the number of publications was observed until 2009 and from 2016 onward; however, no general trends of increase or decrease in production were observed during the entire period.

Articles published over a longer period of time within the coverage period adopted in the systematic search were published by Ghana [28], Japan [102] and the USA [14] in 2003, while the most recent productions were published by India [50,51], Bangladesh [93], China [44], England [98], Iraq [101], Spain [66] and Vietnam [59]. In 2020, the year with the highest number of publications (8 articles) was 2009, followed by 2018 (7 articles), 2015 and 2011 (6 articles), and 2019, 2008 and 2007 (5 articles). There were no scientific productions selected for this review in the period between 2000 and 2002.

### 3.2. Sampling and Studied Species

Different frequencies were observed regarding origin, habitat, fish species and environmental samples collected as units of analysis in the articles. In general, these variations occurred due to the different objectives carried out in research on *Salmonella* spp. regarding microbiological investigation and safety of these foods.

Most of the articles collected samples from aquaculture (54 articles) developed in fresh water (44 articles), indicating greater interest in scientific research in the surveillance of *Salmonella* spp. about this aquaculture sector.

In total, 192 animal species were investigated in the studies included here, including 136 representatives of fish (76 species identified), 20 representatives of shrimp (8 species identified) and 36 representatives of bivalve mollusks, other crustaceans and seafood (18 species identified). Therefore, fish was the category of aquaculture species that was most used for investigations of *Salmonella* spp. in the selected articles, appearing in at least 58 articles, 31 of which investigated only fish samples. Shrimp was present in 12 articles, while other species of crustaceans and bivalve mollusks were present in 16 articles. Figure 8 demonstrates some of the most reported species in the articles.

Among the fish species are *Oreochromis niloticus* [23], *Cyprinus carpio* [101], *Salmo salar* [82,106], *Sparus aurata* [62], *Catla catla* [44], *Colossoma macropomum* [72], *Rachycentron canadum* [53], *Dicentrarchus labrax* [66], *Trachurus trachurus* [69] and *Oncorhynchus mykiss* [69]. The shrimp species were *Litopenaeus vannamei* [58], *Paeneus monodon* and *P. vannamei* [61]. Other species of crustaceans and bivalve mollusks reported were *Mytilus edulis* [71,97], *Midye Dolma* [92], *M. galloprovincialis*, *Venerupis pullastra*, *Ruditapes philippinarum*, *Dosinia exolete* and *Cerastoderma* sp. [27]. Fewer scientific articles have focused on *Salmonella* spp. only in environmental samples of water [25,74,83,109] and sediments [108]. A description of the animal species investigated in the selected scientific articles can be found in Appendix A.

### 3.3. Research Methodologies and Analyzed Aliquots

All studies used microbiological culture methodologies for the investigation and isolation of *Salmonella* spp., of which 21 used the PCR technique and 1 used the qPCR technique [25] concomitantly with culture.

In general, in the diagnostic techniques applied, tests were performed to determine colony characteristics, morphology of the isolates, Gram stain reaction, indole test, methyl red and Voges–Proskauer tests, use of citrate, oxidase test, cell motility, catalase, hydrogen sulfide production, sugar utilization, nitrate reduction, gelatin hydrolysis, starch hydrolysis and test reading.

Some of the studies (17) reported the presence of *Salmonella* spp. according to standard method ISO 6579 (pre-enrichment in buffered peptone water, incubation at 41.5 °C for 24 h in BOD, enrichment in Rappaport–Vassiliadis (RVS), and with tetrathionate Muller-Kauffmann Novobiocin (MKTTn), incubated at 42 °C and 37 °C for 24 h in BOD, respectively, followed by isolation for typical red colonies with a black center and translucent with a red halo on xylose lysine deoxycholate (XLD) agar incubated at 37 °C for 24 h and later submitted to biochemical confirmations), while 13 studies reported the method recommended by the FDA’s Bacteriological Analytical Manual.

Other methodologies recommended by important bodies, such as the APHA (American Public Health Association) [13,75,85] and Association of Official Analyst’s Chemists (AOAC) [44,73,104], were also used by researchers in at least six studies.

Regarding the PCR techniques used in twenty-one studies, the main target genes were the *invA* gene [22,27,50,51,52,56,61,75,87] and the *16S rRNA* gene [25,66,88,94,95,102]. Other targets, such as the *rfb* gene [104], *hns* and *invE* [52], and the *fliB-fliA* intergenic region [57], were also used.

The number of samples analyzed was quite heterogeneous. The publication with the lowest number of samples analyzed was 2 portions of fish curry [53], while the study with the highest number was 10,757 samples of water and live bivalve mollusks [68].

It was possible to verify that the articles used one to seven sample units for the microbiological analysis of *Salmonella* spp. (Figure 9), only two articles are above that [14,48]. To improve the presentation in Figure 9, the sampling units were categorized into “environment” when the articles performed microbiological analyses for *Salmonella* spp. In water, ice and sediment samples [9,17,25,28,66,85], “body” for the analysis of body parts such as shell, head, prawns, carapaces, gills, skin, mucus and surface swabs [43,56,59], “viscera” for the analysis of liver, kidneys, spleen, intestine, hepatopancreas, GI tissue [57,93,102,106], “tissue” for analysis of muscle, fillet (fresh, frozen, salted, smoked and vacuum-packed), meat, meat batter, carcass, blood and brain [70,72,73,82], “biofloc” [96] and “feces” [18,51]. A description of the sample rates analyzed in the selected scientific articles can be found in Appendix A.

### 3.4. Salmonella spp. Positive Reported

Detection of *Salmonella* spp. Was positive in 56 (70%) articles, while the remaining 24 (30%) were not detected. Among the positive samples for the presence of *Salmonella* spp., the lowest percentages of contamination detected were 0.93% and 1.43%, representing positivity for only a single sample [13,99]. The highest number of positives detected was 217 isolates (29.7%) [14]. Most of the “nondetectable” results are in investigations that use only one sample unit (14 articles).

The number of *Salmonella* serotypes identified in scientific articles was also heterogeneous (Figure 10), but some of the products (21 articles) reported isolation up to the level of gender. Only one *Salmonella* serotype was identified and reported in some scientific articles, such as *S*. Dublin [13], *S*. Enteritidis [89], *S*. Saintpaul [77], *S*. Senftenberg [65], *S*. Typhimurium [80], *S*. Corvallis [54] and *S*. Infantis [20], while the largest number of serotypes identified and reported in the same article was sixty-four [79]. The most prevalent serotypes found were *S*. Typhimurium investigated in fish and shrimp muscle [80,102], fish gills and intestine [56,86] and *S*. Weltevreden investigated in shrimp muscle and viscera [58], fish muscle [49,79], seafood [14,48], and shrimp and clam muscle [79]. A description of the main serotypes reported in the selected scientific articles can be found in Appendix A.

## 4. Discussion

Salmonellosis is a public health concern on a global scale. Through this integrative review, it was possible to demonstrate the continuous surveillance of *Salmonella* spp. in the aquaculture sector through the analyzed publications from different producing countries located on different continents over the years between 2000 and 2020 (Figure 6 and Figure 7).

In total, 80 articles were included in this study after eliminating those based on eligibility criteria, where the most representative countries in number of productions were Brazil, China, and India. However, not only countries considered to be major world aquaculture producers—such as China [17,43,45], Bangladesh [93], Chile [81], Egypt [83], India [49,51], Norway [106] and Vietnam [57] or even Brazil [18,72,74], which is in production expansion—but also countries with less relevance in activity have carried out surveillance of *Salmonella* spp. This pathogen has a negative impact not only on public health, since salmonellosis can be transmitted through contaminated fish products, but also on the production rates of flocks, and it acts as an important sanitary barrier to commercial transactions between countries.

Emphasizing the importance and surveillance interest of *Salmonella* spp., the scientific productions had their results published in several titles of journals covering different areas of knowledge—it is an essentially multidisciplinary theme that can cover different areas related to health [42,61] and the environment [22,25,98], such as microbiology [47], animal production [9,67] and food science [13,72].

From the textual analysis of the most frequent words and analysis of similarity, it can be inferred, in general, that the studies selected in this integrative review present references that are inherent to the diagnosis of pathogens of importance in public health in aquaculture production—this helps to understand how to maintain vigilance when reporting the microbiological diagnosis of *Salmonella* spp. in aquaculture species, where fish from production stood out. They also reveal other complementary aspects that help to understand the subject more broadly. Among them is the connection that the works make with microbiological diagnosis, encompassing other bacterial pathogens of the Enterobacteriaceae family and other bacterial pathogens such as *Listeria monocytogenes*, also of public health interest, as they can also be spread through water and cause disease transmitted by contaminated food (DTAs—this shows that different categories of fish can also harbor *Salmonella* spp. and, therefore, can increase the risk of salmonellosis outbreaks in humans; *Salmonella* isolates have already been tested for their antimicrobial resistance profile due to the consideration that they are a threat to health.

More than half of the scientific products analyzed reported the presence of *Salmonella* spp. Over the years (between 2000 and 2020), classic microbiological culture techniques have remained fundamental tools in the microbiological diagnosis of *Salmonella* spp. in fish [22,58,63]. These data suggest that the isolation method by culture remains widespread, as the analysis steps provide a combination of factors favorable to the isolation of viable cells of *Salmonella* spp., which can be serotyped, cultured and classified according to the characteristics of virulence, such as biofilm formation capacity and antimicrobial resistance and susceptibility profiles.

The use of complementary methodologies such as molecular diagnosis by conventional [14] or real-time polymerase chain reaction (PCR) [25] to identify *Salmonella* spp., techniques such as MALDI-TOF-mass spectrometry [17,21] and MLST (Multilocus Sequence Typing) for serotyping were conducted to a lesser extent in the studies. Although molecular diagnosis by PCR is considered a more sensitive technique with the advantages of not generating false positive or false negative results [110] and having a diagnosis in a shorter time compared to microbiological techniques, there is the limitation of not being able to differentiate the cells in the diagnosis of viable specimens from the killing of a pathogen for further microbial isolation. Shabarinath et al. [52] investigated 100 samples and detected *Salmonella* spp. in 20% of them using the conventional microbiology technique, while the PCR technique detected positivity in 52%; in contrast, Hollmann et al. [110] performed a microbiological diagnosis technique after a direct PCR examination of the samples, isolating approximately 74% of the positives.

*Salmonella* is one among many other microbiological risks to be avoided in aquaculture production [76,86]; therefore, as it is an important pathogen in public health, it was possible to observe the inclusion of *Salmonella* spp. in microbiological investigations in aquaculture [80,104], since this bacterial genus is not part of the natural microbiota of fish, and there is a lack of studies that can conclude that fish are affected by clinical symptoms that characterize infection. Therefore, even being a pathogen of the Enterobacteriaceae family, it has been possible to identify that *Salmonella* spp. can survive and multiply in places other than the gastrointestinal tract of fish [18,66]. In this sense, *Salmonella* isolates have been detected in fresh or saltwater collections [22,94] in different external anatomical parts, such as mucus, skin, and gills [17,28,86] of fish and imported frozen food products [80]. In addition to showing that these animals are potential reservoirs of *Salmonella*, all these findings serve as a basic tool for control or prevention measures to be taken, to avoid dissemination and cross-contamination in breeding and production systems and consequently to prevent the risk of occurrence of salmonellosis outbreaks in humans.

The studies presented several choices for analysis units of samples from different categories of aquaculture species for the microbiological diagnosis of *Salmonella* spp. This lack of uniformity could be explained by the difference in the different objectives of the production or other contextual factors in the country that could impact the microbial contamination of products from aquaculture to be kept under surveillance. There are also geographic differences in the study sites. Factors related to the study design: type of sample used (muscle or several other parts) and type of tests used to measure bacterial prevalence (microbiology, PCR) or concentration are likely to contribute to the observed differences.

The different ranges of samples and prevalence results may be because most of the fish sampled were collected in a lightly processed form from a farm or market, where the chances of contamination are high and can contribute to many results observed in the studies.

In the 56 studies that reported positive *Salmonella*, 164 serotypes were identified (Figure 10), with 31 of them cited in at least four articles or more. Extrinsic factors, such as region, environment, species of production and types of samples analyzed in the investigations, are some of the factors that contribute to the diversity of serovars.

*S.* Typhimurium was the most frequently identified in 17 scientific articles from different countries on the Asian continent [43,49,56,57,61,92,102], North and South American countries [74,80], and African country [86], and has been isolated from freshwater [74], muscle, intestine and gills of freshwater fish [49,86], marine fish [102] and other seafood [14,48]. It is the serotype that most commonly causes salmonellosis among the nontyphoid serotypes [29] and has been reported as the dominant serovar causing human infection in China [111]. It can also be isolated from animals from different livestock sectors, such as swine, cattle and chickens [112]. These results agree with the analyses performed by Ferrari et al. [5], in which the *S*. Typhimurium serotype presented a cosmopolitan profile, being the most prevalent and disseminated worldwide and being considered an example of a generalist serotype by several food matrices (beef, pork, chicken and fish), but mainly by pork. In recent research, Wang et al. [111] reported that *S*. Enteritidis, Derby, Typhimurium, Thompsom and Aberdeen were the most common serovars detected in chickens, pigs, ducks, aquatic products and turtles, respectively.

*S.* Weltevreden was the second most reported serotype in scientific articles from countries in Asia [17,49,57,61] and North America [79], being closely associated with the aquaculture of freshwater fish [17], marine shrimp [58,61] and various seafood imported into the US [14,79]. This serotype displays global importance in seafood and is most prevalent in South and Southeast Asia [5]; however, in a recent study using over 35,000 *Salmonella enterica* isolates to explore the temporal and spatial dynamics of the dominant serovars in China, the authors found that *S.* Typhimurium is the dominant serovar [111]. To a lesser extent, this serotype has been the cause of outbreaks in North America [4]. Ferrari and others. [5] suggested that fish imports from Asia may also have imported this pathogen, as in their analysis, there was no evidence that *S.* Weltevreden was native to North America. In the present review, two articles with samples of seafood imported into the US reporting the presence of *S.* Weltevreden were selected [14,79]. This is a classic example of the transmission of pathogens from distant regions, forcing the improvement of hygienic-sanitary control measures.

*S.* Newport was also the second most reported serotype in scientific articles from China [45,46], India [48,52] and the USA [14] in seafood samples. Its highest prevalence was detected in North America, being a pathogen transmissible to humans mainly through the consumption of seafood [5]. However, Zhao et al. [14] reported detecting *S.* Newport from various seafood imported from 38 countries.

Other reported serotypes, such as *S*. Stanley (8), *S*. Kentucky and *S*. Hadar (5) and *S*. Heidelberg (2), have also been previously discussed in FSA and NARMS reports as being important in causing salmonellosis in humans [113,114].

All serotypes need to be kept under surveillance, as aquaculture products have become potential sources for the spread of *Salmonella* spp. During the entire period that comprises production (rearing, capture or removal, processing and retailing phases), fish products are subject to contamination by pathogenic microorganisms naturally present in the aquatic environment and other opportunists introduced through animal and human waste during processing and/or preparation of the production chain [7].

## 5. Final Considerations

Descriptive analyses on the mined data of scientific articles made it possible to generate information from a qualitative and quantitative perspective on the microbiological diagnoses of *Salmonella* spp. in aquaculture during the selected years. It was possible to verify a wide variety of journals in which the articles were published, with the area of food science being the topic with the most concentrated articles. The textual analyses of the abstracts organized and classified five main themes that correlate with *Salmonella* spp. in aquaculture connected to topics such as food safety and public health; however, the absence of words or a set of terms obtained in the word cloud and in the analysis of similarity and descending hierarchical classification (DHC) which refer to measures to prevent or control the contamination of fish was observed. Apparently, the studies included were more focused on characterizing the problem (investigation of the occurrence, aquaculture species sampled, identification and characterization of the profile of the isolates) than on the search for strategies to mitigate the risks of contamination in aquaculture. In general, it was possible to observe from the number of annual publications that there were no increasing or decreasing trends in the surveillance of *Salmonella* spp. in aquaculture over the years; however, there is a constant number of publications on the subject. In this sense, interest in monitoring *Salmonella* spp. by classical microbiological diagnosis was shown worldwide, with articles published in several countries in almost all continents (there were no articles selected from Oceania countries). Asian countries are considered the largest aquaculture producers in the world and were the nations that had the most scientific articles selected by this review. As a fish product that is less processed and more subject to microbiological contamination, it was observed that most of the sampling came from aquaculture activities. Several types of fish categories and environmental samples were analyzed by the articles, but fish was the most sampled species in the studies. *Salmonella* spp. was detected from different units of analysis, and from different anatomical parts of the fish in samples of water from aquaculture cultivation and sediments, suggesting that the pathogen adapts to several favorable environments for multiplication and, consequently, contamination. The conventional microbiological diagnostic techniques used in the articles, sometimes supported by PCR, were unanimous and have shown efficiency in terms of surveillance. Most of the isolates detected were identified up to the genus level; however, it was possible to verify many reported serotypes, which helped to better understand the epidemiological process in aquaculture.

Our research did not include *Salmonella* genotypic characterization in the eligibility criteria. We encourage new reviews focusing on the molecular characteristics of *Salmonella* to be carried out.

Finally, the results obtained here can contribute to the promotion of new studies that investigate strategies for the control and prevention of *Salmonella* spp. in fish production, as the need to increase studies with this focus was observed.

## Figures and Tables

**Figure 1 animals-13-00027-f001:**
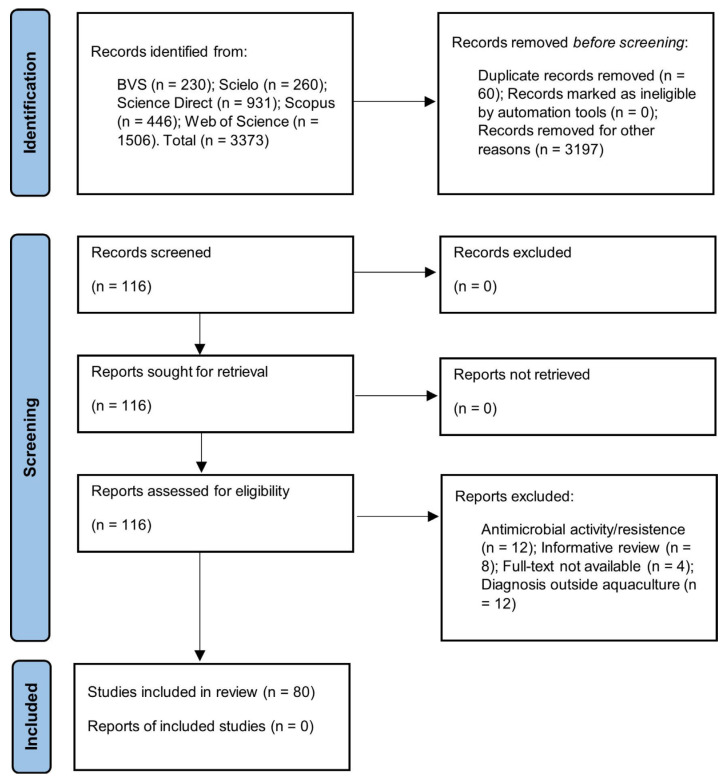
Strategy for the selection of eligible articles.

**Figure 2 animals-13-00027-f002:**
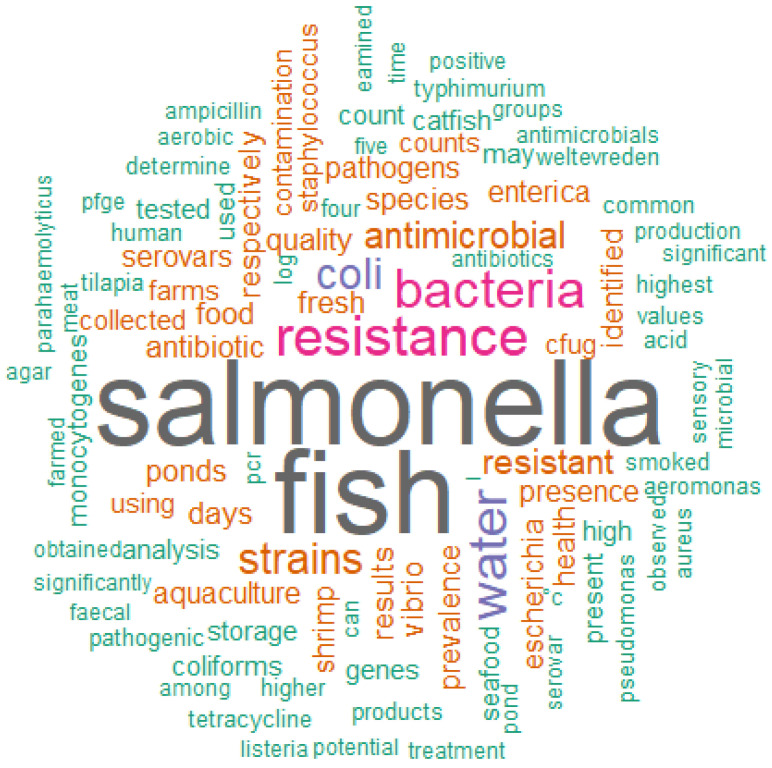
Word cloud formed from the abstracts of the articles selected by the integrative review on microbiological diagnoses of *Salmonella* spp. in aquaculture between 2000 and 2020. N total of words = 17142.

**Figure 3 animals-13-00027-f003:**
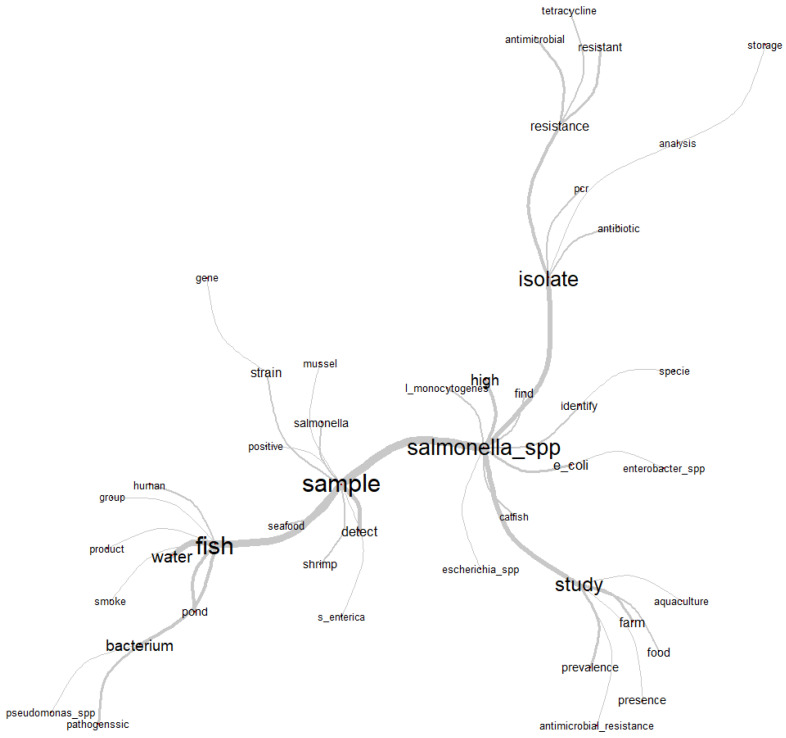
Similarity chart with the relationship of the most used words in the abstracts of the articles selected by the integrative review on microbiological diagnoses of *Salmonella* spp. in aquaculture between 2000 and 2020.

**Figure 4 animals-13-00027-f004:**
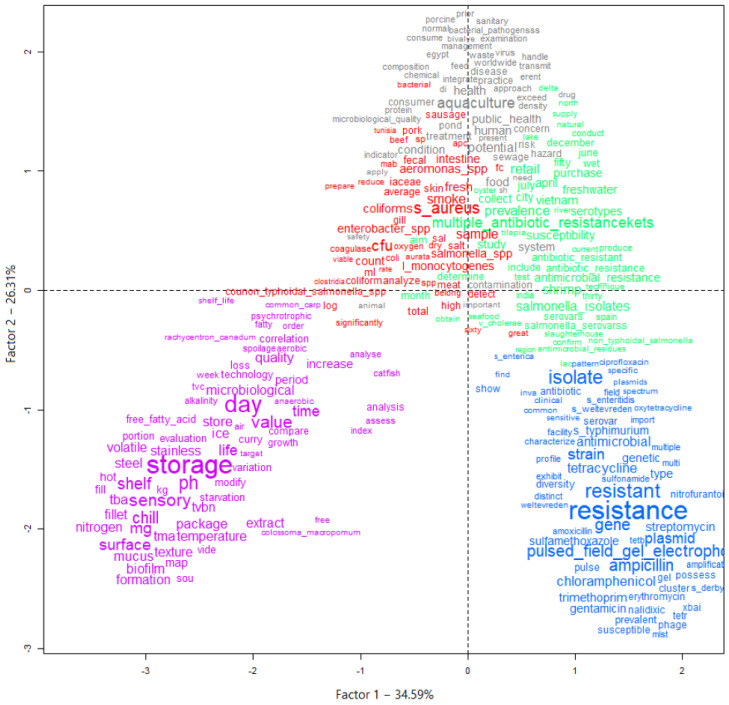
Graph generated by the descending hierarchical classification technique (DHC) containing the textual corpus of abstracts of articles selected by the integrative review on microbiological diagnoses of *Salmonella* spp. in aquaculture between 2000 and 2020, divided by groupings of text segments according to the similarity of terms. Legend: Class 1 = Microorganisms (red); Class 2 = One health (gray); Class 3 = Profiles (green); Class 4 = Antimicrobials (blue); Class 5 = Food safety (purple).

**Figure 5 animals-13-00027-f005:**
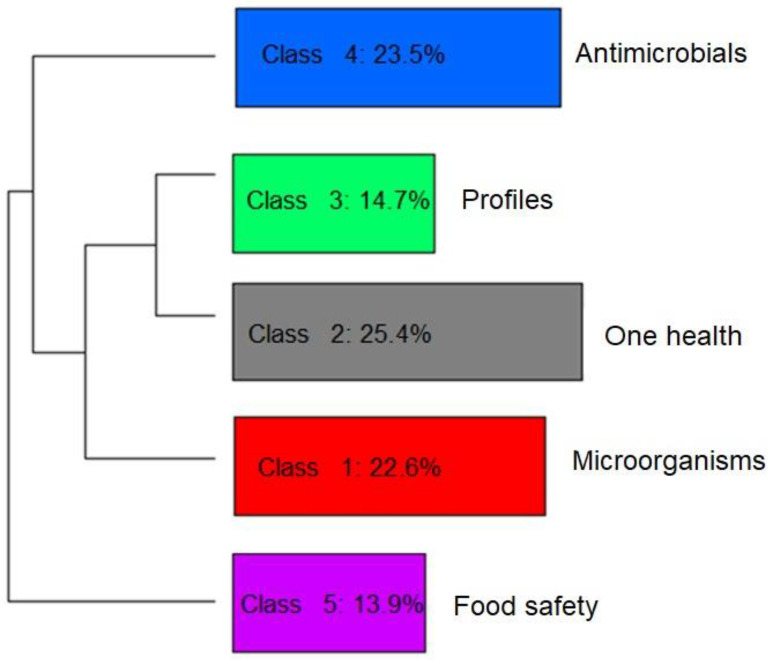
Hierarchical representation and classification of the five groups formed by dividing the textual corpus of the abstracts of the articles selected by the integrative review on microbiological diagnoses of *Salmonella* spp. in aquaculture between 2000 and 2020, according to the similarity of terms.

**Figure 6 animals-13-00027-f006:**
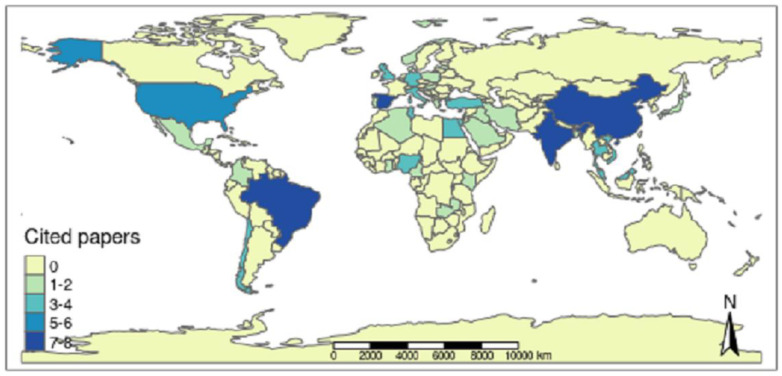
Frequency of scientific articles selected in the integrative review on microbiological diagnoses of *Salmonella* spp. in aquaculture between 2000 and 2020 by countries in different continents.

**Figure 7 animals-13-00027-f007:**
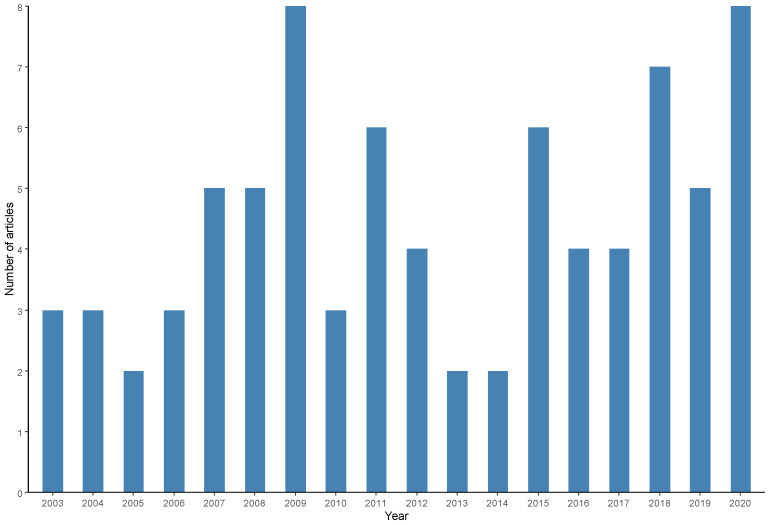
Frequency of scientific articles on microbiological diagnoses of *Salmonella* spp. in aquaculture between 2000 and 2020 by year of publication.

**Figure 8 animals-13-00027-f008:**
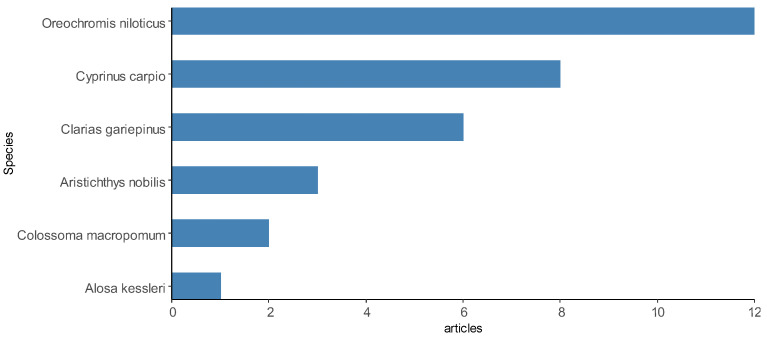
Some aquaculture species most sampled by the articles selected in the integrative review on the microbiological diagnoses of *Salmonella* spp. in aquaculture between 2000 and 2020.

**Figure 9 animals-13-00027-f009:**
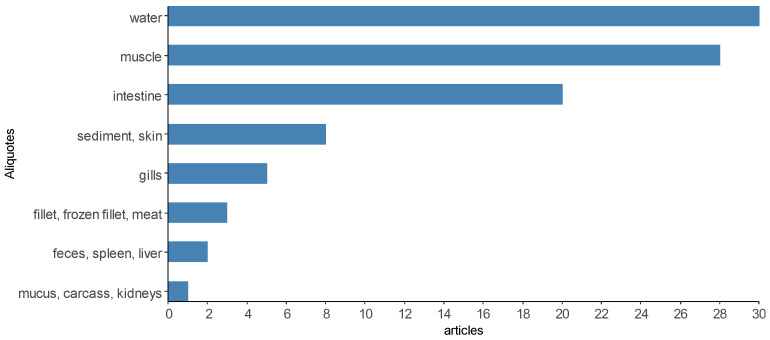
Most analyzed sample aliquots in the scientific articles selected by the integrative review on the microbiological diagnoses of *Salmonella* spp. In aquaculture between 2000 and 2020.

**Figure 10 animals-13-00027-f010:**
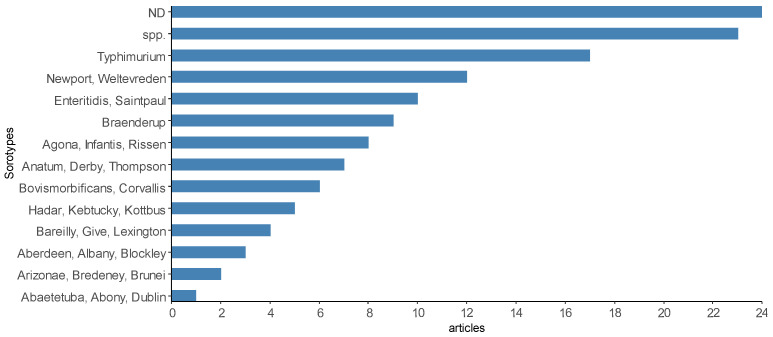
Some of the most reported *Salmonella* serotypes by the scientific articles selected in the integrative review on the microbiological diagnoses of *Salmonella* spp. in aquaculture between 2000 and 2020. Legend: “ND”—undetected *Salmonella*; “spp.”—*Salmonella* spp. reported.

## Data Availability

Not applicable.

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
