# Peer review of "Salmonella spp. in Aquaculture: An Exploratory Analysis (Integrative Review) of Microbiological Diagnoses between 2000 and 2020"

_animals, 2022, doi:10.3390/ani13010027_

Round 1
Reviewer 1 Report
The manuscript "Salmonella spp. in aquaculture: an exploratory analysis of the integrative review of microbiological diagnoses between 2000 and 2020" (manuscript ID: animals-2085568) represents a review on the isolation of Salmonella strains from aquaculture products based on the analysis of the articles present on the topic in the period 2000-2020.
The article is well written and very well structured, for this I congratulate the authors. I list some small changes that I recommend to make before the publication of the manuscript:
- Line 81: change "it can be" with "is".
- Lines 81-87: in the paragraph the presence of Salmonella in different matrices with the relative transmission routes is well discussed. However, I would recommend adding the possible transmission of Salmonella through pets, both mammals and other species (reptiles, fish). I list some articles that could be useful for this purpose:
Dróżdż M, MaÅ‚aszczuk M, Paluch E, Pawlak A. Zoonotic potential and prevalence of Salmonella serovars isolated from pets. Infect Ecol Epidemiol. 2021 Sep 8;11(1):1975530. doi: 10.1080/20008686.2021
Levings RS, Lightfoot D, Hall RM, Djordjevic SP. Aquariums as reservoirs for multidrug-resistant Salmonella Paratyphi B. Emerg Infect Dis. 2006 Mar;12(3):507-10. doi: 10.3201/eid1203.051085.
Meletiadis A, Biolatti C, Mugetti D, Zaccaria T, Cipriani R, Pitti M, Decastelli L, Cimino F, Dondo A, Maurella C, Bozzetta E, Acutis PL. Surveys on Exposure to Reptile-Associated Salmonellosis (RAS) in the Piedmont Region-Italy. Animals (Basel). 2022 Apr 1;12(7):906. doi: 10.3390/ani12070906.
- Line 169: delete "identified".
- Line 397: change "Littopenaeus" with "Litopenaeus".
- Line 398: delete "L." after "edulis".
- Line 399: change "Mytilus galloprovincialis" with M. galloprovincialis".
- Line 400: add "sp." after "Cerastoderma".
- Line 517: add the complete name of MISLT".
- Line 596: delete Salmonella spp.""
Author Response
Dear reviewer
Thank you for the useful comments and suggestions on the structure of our manuscript. We modified the manuscript accordingly, and the detailed corrections are listed below point by point:
Observations: The alterations and adaptations requested by the reviewers are marked up using the “Track Changes” function and new text are in highlight in the main text.

Reviewer 2 Report
The manuscript entitled “Salmonella spp. in aquaculture: an exploratory analysis of the integrative review of microbiological diagnoses between 2000 and 2020“performed a systematic integrative review of the microbiological diagnoses of Salmonella spp. in aquaculture between 2000 and 2020 with the objective of characterizing and contributing to the promotion of measures to control and prevent this pathogen in aquaculture production. The authors selected studies were published between 2000 and 2020, on samples of aquaculture animal production (fish, shrimp, bivalve mollusks, and other crustaceans) and environmental samples of aquaculture activity (farming water, soil, and sediments). Applied methods are scientifically sound.
Overall this work will contribute to the global understanding of Salmonella in aquaculture and the emerging Salmonella clones that pose public health threats and food safety.
Minor:
1. However, It would be better to provide an analysis result of genomic characteristics.
2. Line 56: The most prevalent Salmonella serovars reported were S. Typhimurium, S. Weltevreden and S. Newport? How to understand the origin and distribution of these serotypes? In freshwater fish or marine fish? In the recent paper, reported by Wang and their colleagues Salmonella Aberdeen and Thompson was the most common serovar, respectively. (https://doi.org/10.1093/nsr/nwac269, The temporal dynamics of antimicrobial-resistant-Salmonella enterica and predominant serovars in China) The author should discuss this difference in the manuscript.
3. Please provide a table summary the reference (for example, Isolation origin, time, region, etc) the authors used to the manuscript.
Author Response

(The authors gave the same response as above.)
